https://doi.org/10.1038/s41467-022-28886-5　　**OPEN**

# A zirconium metal-organic framework with SOC topological net for catalytic peptide bond hydrolysis

Sujing Wang [1,2,6✉], Hong Giang T. Ly[3,4,6], Mohammad Wahiduzzaman [5,6], Charlotte Simms[3], Iurii Dovgaliuk [1], Antoine Tissot[1], Guillaume Maurin[5], Tatjana N. Parac-Vogt [3✉] & Christian Serre [1✉]

The discovery of nanozymes for selective fragmentation of proteins would boost the emerging areas of modern proteomics, however, the development of efficient and reusable artificial catalysts for peptide bond hydrolysis is challenging. Here we report the catalytic properties of a zirconium metal-organic framework, MIP-201, in promoting peptide bond hydrolysis in a simple dipeptide, as well as in horse-heart myoglobin (Mb) protein that consists of 153 amino acids. We demonstrate that MIP-201 features excellent catalytic activity and selectivity, good tolerance toward reaction conditions covering a wide range of pH values, and importantly, exceptional recycling ability associated with easy regeneration process. Taking into account the catalytic performance of MIP-201 and its other advantages such as 6-connected $Zr_6$ cluster active sites, the green, scalable and cost-effective synthesis, and good chemical and architectural stability, our findings suggest that MIP-201 may be a promising and practical alternative to commercially available catalysts for peptide bond hydrolysis.

[1] Institut des Matériaux Poreux de Paris, Ecole Normale Supérieure, ESPCI Paris, CNRS, PSL Université, Paris, France. [2] CAS Key Laboratory of Microscale Magnetic Resonance and Suzhou Institute for Advanced Research, University of Science and Technology of China, Hefei, China. [3] Laboratory of Bioinorganic Chemistry, Department of Chemistry, KU Leuven, Leuven, Belgium. [4] Department of Chemistry, College of Natural Sciences, Can Tho University, Can Tho, Vietnam. [5] ICGM, Univ. Montpellier, CNRS, ENSCM, Montpellier, France. [6] These authors contributed equally: Sujing Wang, Hong Giang T. Ly, Mohammad Wahiduzzaman. ✉email: sjwang4@ustc.edu.cn; tatjana.vogt@kuleuven.be; christian.serre@ens.psl.eu

A peptide bond is a type of robust amide bond that connects amino acid residues in proteins, thus having essential importance in biological systems. The remarkable stability of peptide bonds under physiological conditions (with an estimated half-life of 350–600 years at 25 °C in neutral pH conditions) guarantees the intactness of the primary sequence of a protein, but results in a considerable challenge when there is a need to break the bonds[1].

Catalytic hydrolysis is an efficient way of breaking peptide bonds and results in the release of the carboxylic and amine functional groups. In biological systems, this reaction is carried out by enzymes with extremely high reaction rates. Outside of biological systems, the hydrolysis of proteins is an important procedure in areas such as protein-structure analysis, protein engineering, and protein-cleaving drug design[2–4]. Therefore, the high costs and the extreme sensitivity of enzymes to the reaction conditions motivated the development of artificial proteases[2,3], i.e., to achieve adequate reactivity and specificity, as it is very challenging for artificial catalysts to match the exceptional catalytic power of natural enzymes.

Among existing alternatives, homogeneous catalysts based on Lewis acid metal salts often suffer from formation of gels at neutral and basic conditions, leading to loss of reactivity and difficulty in the separating products and reactants[5]. In comparison, metal complexes prevent the formation of gels during catalysis, but have limited reactivity window, toxicity, and poor recyclability[2,5]. More recently, metal-substituted polyoxometalate (POM) clusters were developed as homogeneous catalysts for peptide-bond hydrolysis. In particular, zirconium(IV)-substituted POMs have been reported to achieve an encouraging combination of activity and selectivity in mildly acidic and neutral media[6–8], but their structural dynamic under the reaction conditions and their high solubility make catalyst recycling and product purification problematic. In order to circumvent these shortcomings, insoluble $Zr_6$-oxo-cluster-based metal–organic frameworks (MOFs) were evaluated as state-of-the-art heterogeneous catalysts for accelerating peptide-bond hydrolysis[5,9,10]. The large-pore MOF-808 and NU-1000 (NU, Northwestern University) displayed much better reactivity and recyclability compared with POMs under neutral and mildly acidic conditions, but still suffer from limited stability under alkaline reaction conditions[11,12]. The highly defective UiO-66 (UiO, University of Oslo) and its functional derivatives, on the other hand, showed comparable catalytic performance with POMs but with improved stability under neutral and mildly alkaline conditions. Therefore, it remains a great challenge to develop an efficient heterogeneous catalyst that is effective under various conditions and within a wide range of pH values, while preserving a good catalytic activity and specificity for the catalytic peptide-bond hydrolysis.

Herein, we present the first Zr–MOF with square–octahedron (soc) topological net constructed from $Zr_6$-oxo cluster secondary building units (SBUs) and a tetracarboxylate linker (3,3,5,5′-tetracarboxydiphenylmethane ($H_4$mdip)), denoted as MIP-201 (MIP, materials of the Institute of Porous Materials from Paris), that addresses the challenge of developing a highly efficient and robust heterogeneous catalyst for peptide-bond hydrolysis. MIP-201 possesses catalytically active 6-connected $Zr_6$-oxo-cluster building units in its robust three-dimensional (3D) microporous network, which leads to excellent heterogeneous catalytic performance in accelerating peptide bond hydrolysis under a wide pH range of conditions (acidic, neutral, and basic), a good catalytic activity and selectivity, and a superior catalyst recycling ability. The combined advantage of its catalytic performance, cost-efficient, green and scalable synthesis, and excellent architectural and chemical stability make MIP-201 one of the most promising artificial proteases discovered so far.

## Results

**Synthesis and crystal structure of MIP-201**. $Zr_6$-oxo cluster is the dominant building unit in the fabrication of Zr-carboxylate MOFs. The connection numbers of $Zr_6$-oxo clusters cover a wide range of 12, 10, 9, 8, 6, 5, and 4 in the reported examples, showing an extraordinary flexibility in connecting various linkers to generate a large library of different structures[13–16]. While the stabilities of Zr–MOFs constructed with low connection-number (5 and 4) nodes are still under debate[15–18], 6-connection is usually considered as the limit to keep the material stability at an acceptable level[19–21]. Unlike the large number of Zr–MOFs based on high connection-number (above 8 till 12) building units, very few Zr–MOFs are constructed from 6-connected $Zr_6$-oxo clusters, including PCN-224[22], PCN-777[23] (PCN, Porous Coordination Network), UMCM-309[24] (UMCM, University of Michigan Crystalline Material), the interpenetrated Zr–BTB[25], MOF-808[19], and BUT-108 (BUT, Beijing University of Technology)[26].

6-connected $Zr_6$-oxo clusters in Zr–MOFs show two different configuration modes: the hexagonal planar and trigonal prismatic. The hexagonal planar configuration was only observed in PCN-224, while the other five examples share the trigonal prismatic configuration. The trigonal prismatic configuration of the 6-connected $Zr_6$-oxo cluster is of a particular interest for the design of new topological structure of Zr–MOFs[27], since it has shown similar functions as trimer-building units of trivalent metal ions in the MOF-framework fabrication, such as the case of PCN-777 to be related to the metal(III) oxo-trimer-based MIL-100 and MIL-101 (MIL, Materials of Institut Lavoisier)[23].

In this regard, we analyzed the possibility of replacing trivalent metal-trimer SBUs by 6-connected $Zr_6$-oxo clusters in several types of known MOF structures, including MIL-88[28], soc-MOFs[29], and a series of PCN networks[30]. We found that the soc topological net would be one of the most feasible targets to achieve due to the following properties: (1) all the connection sites on the trimer node are occupied by the carboxylate groups from the tetracarboxylate linker molecules, leading to the same configuration of the trigonal prismatic 6-connected $Zr_6$-oxo cluster; (2) the special shape of the tetracarboxylate linkers with appropriate steric hindrance and the separation between carboxylate groups could efficiently force the linkage and structure extending in the way of generating soc net; (3) the reported soc-MOFs displayed excellent stability even with trivalent metal ions, probably due to the hydrophobicity of the structural frameworks, as well as the considerable steric hindrance around the SBU to weaken the attack from water molecules[31,32]. Regarding the selection of linker for this hypothesis, a tetracarboxylate ligand with an appropriate structural flexibility would be even more beneficial if the larger size and elevated rigidity of $Zr_6$-oxo cluster are taken into consideration in comparison with that of trivalent trimers. To that end, $H_4$mdip was finally selected as the most suitable tetracarboxylate linker for the synthesis of soc-Zr–MOF, not only due to its cost-effective and scalable synthesis, but also as its good structural flexibility to adapt various connection environments in the MOF construction is well documented[33,34].

Following the aforementioned guidance, a highly crystalline product (MIP-201) could be isolated from the solvothermal reaction of $ZrCl_4$ and $H_4$mdip in pure acetic acid at 120 °C. Several attempts to obtain larger product particles suitable for single-crystal X-ray diffraction data collection, including the combination of different Zr(IV) precursors with various solvent mixtures (formic acid, acetic anhydride, water, ethanol, etc.), did not lead to any increase in size of the product (Supplementary Fig. 1). In addition, the good tolerance of MIP-201 toward the synthesis conditions allowed the development of green and scalable routes for further practical applications. After

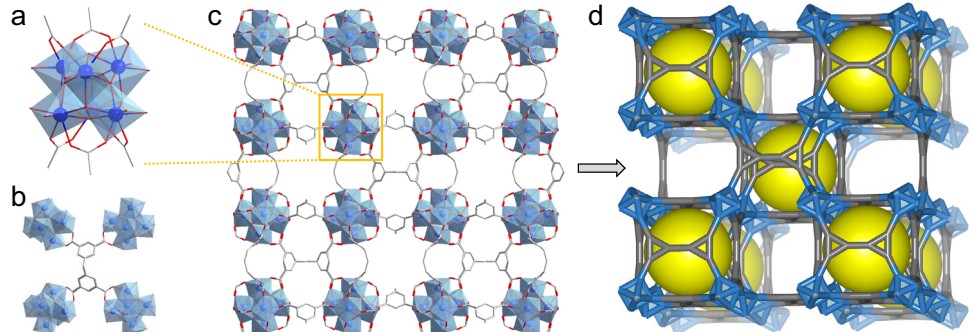

**Fig. 1 Crystal structure of MIP-201. a** The 6-connected $Zr_6$-oxo-cluster building unit with the trigonal prismatic configuration. **b** One tetracarboxylate linker molecule connects the adjacent four $Zr_6$-oxo clusters. **c** The overall structure viewed along the c axis. **d** The square–octahedron (soc) topological net of MIP-201 structure (Zr in blue, C in gray, O in red, and the yellow ball stands for the cavity).

optimization, an all-green reaction of refluxing noncorrosive $Zr(SO_4)_2·4H_2O$ and $H_4mdip$ in the mixture of water and acetic acid under ambient pressure was set for the scale-up synthesis of MIP-201 product at the 10 g scale associated with a good yield (90% based on the linker used).

The crystal structure of MIP-201 was solved by a combined analysis of high-resolution powder X-ray diffraction (PXRD) data and a computational topology-guided reverse engineering approach (Supplementary Figs. 2 and 3, Supplementary Table 1). MIP-201 with a formula of $[Zr_6(\mu_3\text{-}O)_4(\mu_3\text{-}OH)_4(acetate)_{0.24}(OH)_{5.76}(H_2O)_{5.76}(mdip)_{1.5}]$ was found to crystallize in a cubic $Im\text{-}3$ space group (No. 204) with unit-cell parameters of $a = 24.5847(11)$ Å and $V = 14859.2(11)$ Å$^3$. It features a 3D microporous framework composed of 6-connected $Zr_6(\mu_3\text{-}O)_4(\mu_3\text{-}OH)_4$ oxo-cluster SBUs with mdip tetratopic linker molecules for separating and spacing. As shown in Fig. 1a, each $Zr_6$ oxo-cluster SBU comprises six carboxylate groups from six different linkers at the polar positions and six terminal acetate groups or -$OH/H_2O$ pairs locating around the equatorial plane. Taking account of the molecular conformation character of mdip over the other rigid tetratopic linkers, the considerable flexibility of the methylene group that connects two benzene rings plays an important role in this case by forcing the mdip molecule to adapt this connection mode (Fig. 1b). MIP-201 represents, to our knowledge, the first example of this type of 6-connected $Zr_6$ cluster SBU in Zr–MOFs built with tetratopic linkers.

The four carboxylate groups of mdip linker are fully deprotonated to connect eight separated Zr(IV) ions binding four adjacent SBUs together (Fig. 1b). As a result of the high flexibility of the methylene group, mdip molecules are able to adjust their length and width according to the corresponding connection requirements, and thus realize the interconnection of eight neighboring SBUs, giving rise to a distorted cubic pocket with a free diameter around 7 Å. Free voids generated between the neighboring pockets are present, showing an accessible dimension of around 10 Å, leading to a theoretical nitrogen-accessible surface area of $1000\,m^2\,g^{-1}$ and a total free pore volume of $0.50\,cm^3\,g^{-1}$ calculated from the crystal structure (Supplementary Figs. 2 and 4). However, residual $SO_4^{2-}$ groups are still trapped inside the pore of the MOF (S/Zr = 27.6/72.4, atomic ratio), as evidenced by scanning electron microscopy with energy-dispersive X-ray spectroscopy (SEM-EDX) measurement. This resulted in an experimental Brunauer-Emmett–Teller (BET) area of $680\,m^2\,g^{-1}$ and a total pore volume of $0.30\,cm^3\,g^{-1}$ deduced from nitrogen porosimetry, slightly below the theoretical values. The similar observation of strongly trapped $Cl^-$ species in the MOF pore was found when $ZrCl_4$ or $ZrOCl_2·8H_2O$ were used as the reactants for the synthesis of MIP-201, which has been recognized as a general issue for the full activation of Zr–MOFs.

The overall 3D structure of MIP-201 is packed with cubic pockets and free void spaces (Fig. 1c), thus displaying a typical network of $cdj$ topology. It is worthy to note that MIP-201 is, to our knowledge, the first example of soc-type network in Zr–MOFs (Fig. 1d).

**Stability of MIP-201**. In order to ensure the potential of MIP-201 in various applications, thermal and chemical stability tests under different conditions were carried out. Temperature-dependent PXRD and thermogravimetric analysis (TGA) (Supplementary Figs. 5 and 6) indicate that MIP-201 maintains its crystal structure up to 375 °C, which is adequate for most of the applications related. Chemical resistance of MIP-201 was checked under diverse conditions, including boiling water, fuming acid, super-acid, and basic conditions. As shown in Fig. 2a, the long-range order of the MIP-201 structure was maintained very well under all the tested conditions supported by their almost identical PXRD patterns. Long-duration contacts with boiling water, fuming acids, aqua regia, highly concentrated $H_3PO_4$, and basic conditions (pH = 10 buffer and $NH_4OH$ vapor) did not show notable degradation of the crystalline structure of MIP-201. Nitrogen-sorption measurements at 77 K carried out on samples treated with some extremely harsh conditions supported the good reservation of the MOF long-range order and porosity (Fig. 2b), despite having structural defects generated by HCl or aqua-regia treatments. Therefore, MIP-201 displays excellent stability for applications, especially in regard to bio-related applications, which generally require good tolerance toward the presence of phosphate species.

**Catalytic performance of MIP-201 in promoting peptide-bond hydrolysis**. The initial evaluation of the MIP-201 catalytic performance toward peptide-bond hydrolysis was tested using a simple Gly–Gly dipeptide. A mixture of equimolar amounts (2.0 μmol) of Gly–Gly and MIP-201 was incubated at 60 °C and pD 7.4 (Supplementary Fig. 7). The progress of peptide-bond hydrolysis was followed by $^1H$ NMR spectroscopy (Supplementary Figs. 8 and 9). To confirm that the catalytic activity was due to the MIP-201 material, and was not caused by Zr(IV) ions leached in solution, the MIP-201 powder was removed from the reaction mixture after 14 h by centrifugation, and the homogeneous solution was allowed to react further. As seen from Fig. 3a, no additional Gly–Gly hydrolysis was observed after removing MIP-201 from the reaction mixture, indicating that the peptide-bond hydrolysis is exclusively associated with catalytically active sites in the solid MIP-201 material and not caused by Zr(IV) ions or $Zr_6$-oxo clusters that might have leached into solution.

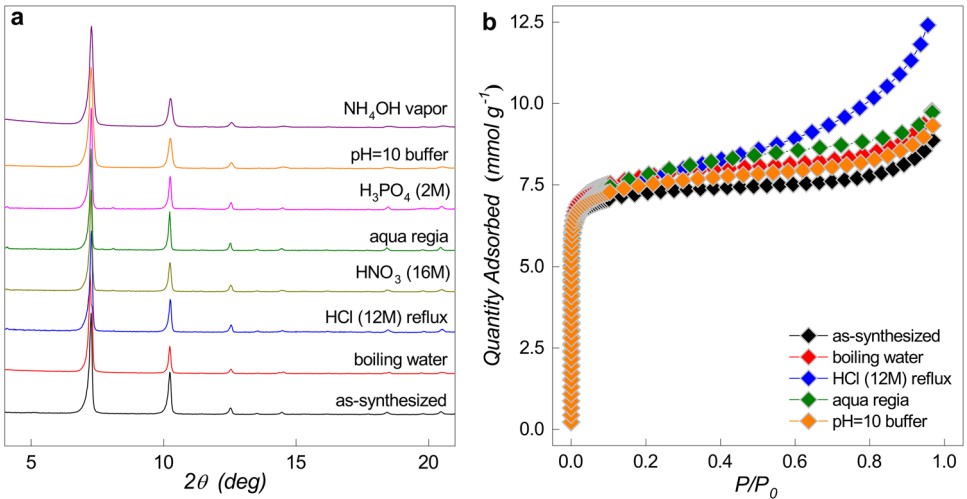

**Fig. 2 Chemical stability of MIP-201. a** PXRD patterns (CuK$_\alpha \approx 1.5406$ Å) of MIP-201 samples treated under various chemical conditions. **b** Nitrogen adsorption isotherms of MIP-201 samples after some typical chemical treatments collected at 77 K (samples were refluxed in water before thermal activation at 120 °C for each condition).

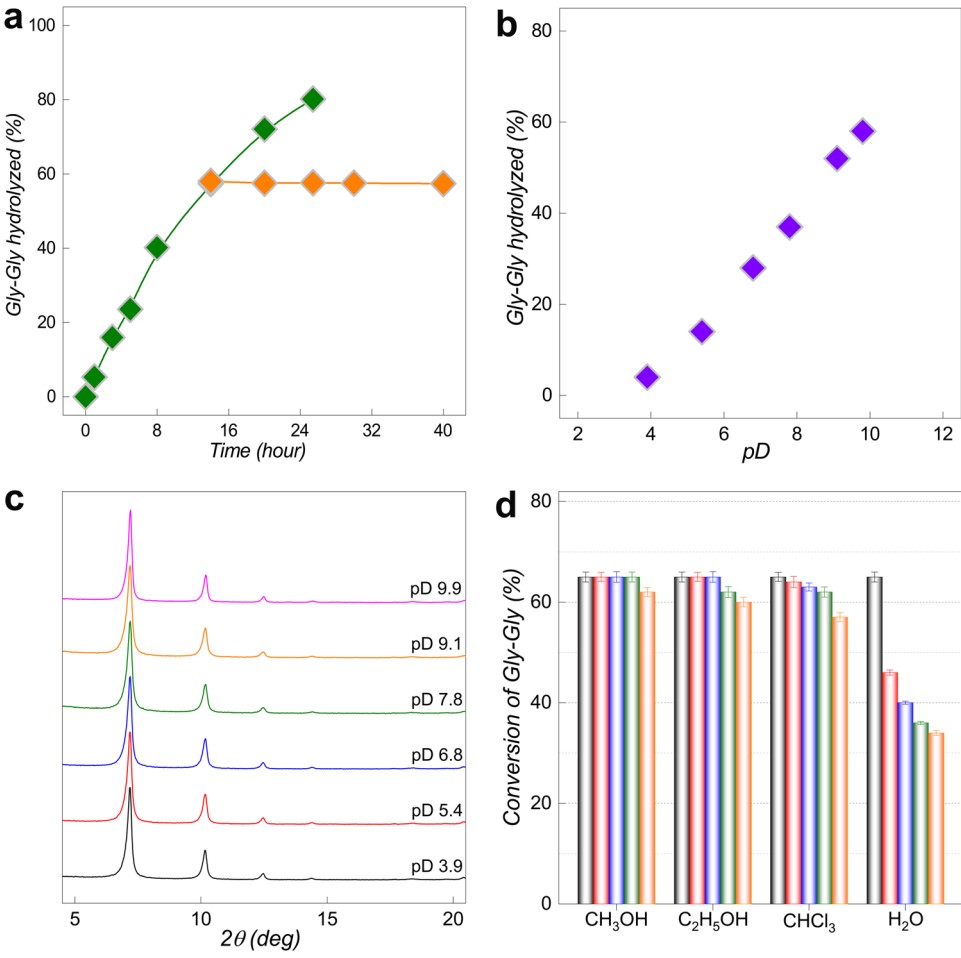

**Fig. 3 Catalytic performance of MIP-201 for hydrolysis of Gly–Gly. a** Hydrolysis of 2.0 μmol of Gly–Gly in the presence of 2.0 μmol of MIP-201 before (green square) and after (orange square) the removal of MIP-201 (pD 7.4 and 60 °C). **b** Conversion of Gly-Gly after eight-hour hydrolysis of Gly–Gly (2.0 μmol) catalysed by MIP-201 (2.0 μmol) at 60 °C and different pD values. **c** PXRD patterns of MIP-201 after eight-hour catalysis at 60 °C in reaction solution of different pD values ($\lambda_{Cu} \approx 1.5406$ Å). **d** Conversion of Gly–Gly after sixteen hours at 60 °C in the presence of MIP-201 for five reaction cycles. Different organic solvents were used to wash and exchange with water before a four-hour activation at 120 °C.

The rate constant of Gly–Gly hydrolysis ($k_{obsd}$) catalyzed by MIP-201 was calculated to be $1.85 \times 10^{-5}\,s^{-1}$ (corresponding to $t_{1/2} = 10.4$ h) at 60 °C and pD 7.4 (Supplementary Fig. 8). This represents an enhancement of nearly two orders of magnitude compared with the reactions catalyzed by Zr(IV)–POMs under the same experimental conditions[6,7], and a 2500-times enhancement compared with the uncatalyzed hydrolysis of Gly–Gly ($k_{obsd} = 7.4 \times 10^{-9}\,s^{-1}$, corresponding to $t_{1/2} = 3$ years). It is noteworthy that the rate of Gly–Gly hydrolysis by MIP-201 is comparable to that previously observed with MOF-808[5] when the larger surface area of MOF-808, which allows for more adsorption of peptide substrate, is taken into consideration (Supplementary Fig. 10). Furthermore, NU-1000, constructed from 8-connected $Zr_6$-oxo SBUs and exhibits mesoporous cavities associated with a large BET area ($2200\,m^2\,g^{-1}$), showed a peptide-bond hydrolysis rate that was more than one order of magnitude slower than that of MIP-201[9]. This highlights the important role of 6-connection of the $Zr_6$-oxo node on the catalytic activity and how critical the active-site density is for the efficiency of a MOF artificial peptidase, since other Zr–MOFs with larger cluster connectivity exhibit lower catalytic activities[9,10]. Detailed density-functional theory (DFT) calculations on the peptide-bond hydrolysis by MOF-808 previously confirmed the central catalytic role of $[Zr_6O_8]$ cluster, and revealed that the reaction is facilitated by peptide coordination to two adjacent Zr(IV) centers in the cluster[35]. Such activation by two Zr(IV) centers likely explains the superior reactivity of Zr–MOF materials with respect to Zr(IV) polyoxometalate complexes, in which only a single Zr(IV) is present for peptide-bond activation[36].

In accordance with this mechanism, the hydrolysis of amide bond in N-methylacetamide by MIP-201 was not observed after 8 days of reaction (Supplementary Fig. 11). This highlights the important role the C-terminal carboxyl group plays in acceleration of the reaction, possibly by enabling binding of the dipeptide to the catalytically active Zr(IV) sites. These Zr(IV) sites are present on the outer surface and in the pores of MIP-201, and in principle, both sites could lead to the activation of Gly–Gly. Previous studies of $N_2$-sorption isotherms with NU-1000 revealed slight reduction in the BET area before and after reaction and ≈15% reduction of the largest pore's diameter after incubation with Gly–Gly, evidencing that the pores are at least partially filled after reaction with the peptide[9].

The catalytic hydrolysis of peptide bond was further investigated as a function of pD. The mixture of Gly–Gly and MIP-201 was incubated at 60 °C in solutions with pH ranging from 3.9 to 9.9 and the corresponding results are presented in Fig. 3b. In the range from acidic to neutral pD, the rate enhancement was observed along with the pD increase. Similar evolution was observed in other catalytic systems, and was explained by the gradual reduction of Gly–Gly protonation that facilitates the effective binding to the catalytically active metal-ion centers[2]. Remarkably, with further pD increase, a very distinct catalytic performance of MIP-201 was observed; reaction rates displayed a steady increase with more than 50% rate enhancement observed at pD = 9.9 compared with physiological pD. On the contrary, decrease in reaction rates has been reported for many other catalysts that were used under alkaline conditions, and was mainly attributed to the instability of the catalysts. For instance, metal salts typically form inactive insoluble gels or solids under mildly basic solutions[5], while POMs are largely unstable under alkaline conditions and show the best catalytic efficiency in slightly acidic situations[6,7]. Similarly, the PXRD patterns of both MOF-808 and NU-1000 revealed crystal-structure degradation and reduced catalytic activities when the pD of the solution was raised above 8.0 (Supplementary Fig. 12)[5,9]. On the contrary, the

stability of MIP-201 within the full pD range of the reaction conditions was confirmed by PXRD (Fig. 3c) and IR spectroscopy (Supplementary Fig. 13), which were carried out on the MIP-201 samples collected after the completion of catalytic reaction. These characterizations indicated that the structure of MIP-201 remains intact with no evidence of degradation under all examined reaction conditions.

The recycling of MIP-201 as a heterogeneous peptidase was further investigated, revealing a remarkably high architectural stability after the direct regeneration by washing with water. Although a decrease of catalytic activity was observed for water-regenerated MIP-201 (from 71% yield observed in the second run to 55% in the fourth run, Fig. 3d), it could be rationalized by the microporous nature of the MIP-201 structure, in which a partial pore blocking may lead to a slight reduction of the number of accessible catalytically active sites. It is worth mentioning that comparable yields were observed in the fifth and fourth cycles, supporting the conclusion that the number of the accessible catalytic sites has approached an equilibrium, while the overall architecture of the MIP-201 framework was maintained during the entire catalytic cycles. A similar observation was found when chloroform ($CHCl_3$) was used to regenerate catalyst after the reaction, while both methanol and ethanol were shown to be excellent solvents for regenerating MIP-201 (in accordance with recyclability studies with other MOFs in general)[5]. In contrast to MIP-201, the recycling of MOF-808 after Gly–Gly hydrolysis indicated a limited architectural stability of the MOF-808 framework, which impeded its regeneration by direct water washing and exchanging. This was attributed to the combined effect of considerable structural defects present in MOF-808 and the high surface tension of water[5,11]. This resulted in a drop of more than 90% in the second run of catalysis after washing with water, which was associated with a partial structural collapse of MOF-808 with the direct hydrolytic treatment.

**Hydrolysis of horse-heart myoglobin (Mb) catalysed by MIP-201.** As the high catalytic activity, good stability, and recycling ability of MIP-201 were demonstrated in the hydrolysis of a model dipeptide, further evaluation of its catalytic performance was carried out on horse-heart myoglobin (Mb), a protein containing 153 amino acids with approximate Mw of 16.95 kDa. Previous studies have shown that water-soluble Zr(IV)-substituted POMs[37] and insoluble $Hf_{18}$ metal–oxo cluster[38] were able to selectively hydrolyze Mb at peptide bonds containing Asp residues. In that respect, Mb represents a good model protein for evaluating and comparing the catalytic potential of MIP-201 to other reported artificial proteases. The reactions were carried out by mixing Mb with MIP-201 in HEPES buffer at 60 °C and analyzing reaction aliquots by SDS-PAGE at different time increments. The selective hydrolysis of Mb was evidenced by the appearance of new bands with lower molecular weights at approximately 14.9, 13.6, 12.5, 11.0, 10.1, and 7.1 kDa (Fig. 4a), and similar SDS-PAGE pattern was observed when pure water was used as medium (Fig. 4b). In the presence of MIP-201, the hydrolysis of Mb could be already observed after one hour of incubation, and the intensity of bands with lower molecular weight (MW) progressively increased with time. The control experiments showed that hydrolysis of Mb was not observed in the absence of MIP-201 after a four-day incubation at 60 °C, confirming the catalytic role of MIP-201 in protein hydrolysis.

The Mb hydrolysis has also been followed at 37 °C, with 34% of the protein hydrolysis observed after 1 h and 54% after 24 h (Supplementary Fig. 14, Supplementary Table 2). However, fewer fragments were observed when Mb hydrolysis was performed at 37 °C. Fragments at 12.5 kDa were the most intense, while fragments

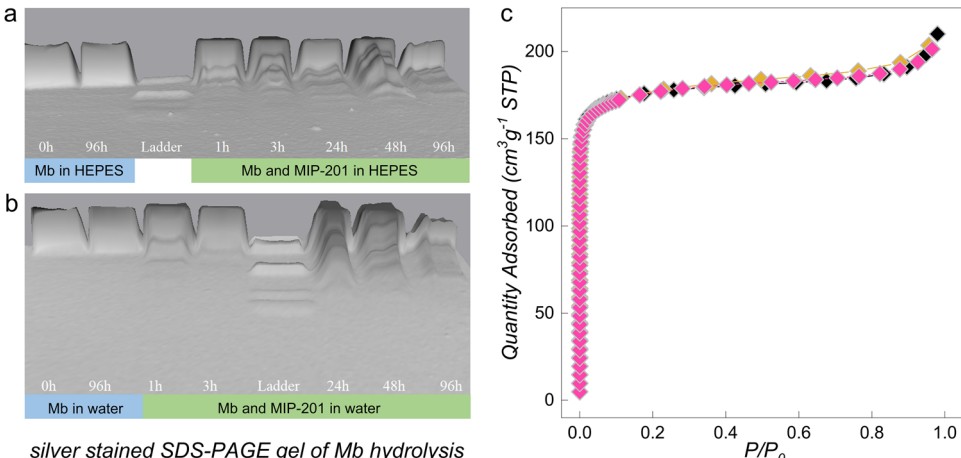

**Fig. 4 Hydrolysis of horse-heart myoglobin (Mb) catalyzed by MIP-201. a** Silver stained SDS-PAGE gel of Mb hydrolysis in the presence of MIP-201 in HEPES buffer (pH 7.4 and at 60 °C). **b** Silver-stained SDS-PAGE gel of Mb hydrolysis in the presence of MIP-201 in water (pH at 7.4 and at 60 °C). **c** Nitrogen adsorption isotherms of MIP-201 samples before (black square) and after the application in hydrolysis of Mb in HEPES buffer (orange square) and water (pink square).

at 11.2 kDa and 10.5 kDa appeared after 3 and 24 h of reaction, respectively. Interestingly, samples incubated at 37 °C for 48 h or longer, did not show any fragments in SDS-PAGE, while the band corresponding to intact Mb also decreased in the intensity, in agreement with a significant adsorption of Mb and its hydrolytic fragments after prolonged incubation times. This suggests that temperature has a significant effect on the Mb hydrolysis by MIP-201, both by influencing the kinetics of hydrolysis and the number of produced fragments (Supplementary Table 3). In addition to facilitating the hydrolysis of kinetically very inert peptide bonds, the elevated reaction temperature also causes unfolding of the protein, which makes peptide bonds in Mb more accessible for interaction with the catalyst. The MWs of the protein fragments formed by hydrolysis of Mb were estimated using MW ladder in SDS-PAGE experiments. While the exact position of the cleavage sites has not been determined at this point, the bands observed at 60 °C give indication that Mb was likely hydrolyzed in the vicinity of Asp20–Ile21 (14.9 kDa), Asp122–Phe123 (13.6 kDa), Asp109–Ala110 (12.5 kda), and Asp60–Leu61 (10.1 kDa), while the bands at 11.0 and 7.1 kDa likely originated from the cleavage of Mb at two sites (at Asp20–Ile21 and Asp122–Phe123 for the band at 11.0 kDa; Asp60–Leu61 and Asp126–Asp127 for 7.1 kDa). At 37 °C, the hydrolysis most likely occurred at Asp109–Ala110 (12.5 kDa), while the bands at 11.2 and 7.1 kDa likely originated from the cleavage of Mb at two sites as described above (at Asp20–Ile21 and Asp122–Phe123 producing the band at 11.2 kDa; Asp60–Leu61 and Asp126–Asp127 for 7.1 kDa). The position of these bands was in accordance with the previously reported affinity of Zr(IV)-based artificial proteases to preferentially cleave proteins next to Asp residues[37]. The hydrolysis of these Asp–X-peptide bonds in Mb has been previously observed in the presence of Zr–POMs; however, those reactions were much slower, with the observable hydrolysis occurring only after 48 h at pH = 5.0[37]. We are currently working on the development of a protocol for mass spectrometric analysis, which will allow unambiguous detection of the cleavage sites.

Nevertheless, the ability of MIP-201 to hydrolyze Asp–X bonds was tested by using Asp–Gly as a model dipeptide. The kinetic experiments showed that the rate of Asp–Gly peptide-bond hydrolysis occurs with the rate constant of $k_{obsd} = 7.20 \times 10^{-6}\,s^{-1}$, which is somewhat slower than the hydrolysis of Gly–Gly (Supplementary Fig. 15). These results are consistent with the previous report about hydrolysis of various dipeptides by MOF-808, where slower hydrolysis of larger dipeptides was explained by the

steric hindrance imposed by bulky side chains, which may hinder the access of the peptide carbonyl oxygen atom to the site and might potentially limit the diffusion of the dipeptide in the pores of MIP-201[5].

The stability of MIP-201 after the reaction with Mb was evaluated by a combination of several techniques performed on samples collected from large-scale reactions. Nitrogen porosimetry (Fig. 4c), PXRD (Supplementary Fig. 16), and FTIR (Supplementary Fig. 17) measurements confirmed that the structure of MIP-201 remained intact after hydrolytic experiments with Mb, further validating its potential as a nanozyme for selective protein hydrolysis. Considering the structure of MIP-201 and the size of Mb protein, it is very likely that the hydrolysis of Mb occurs on the outer surface of the particles. A previous study with NU-1000 revealed very little changes in the MOF pore size upon incubation with the protein, while significant changes in the BET area were detected, indicating significant absorption of protein onto the surface and not in the pores of MOF[9].

We demonstrate in this work that the cubic microporous MIP-201, the first Zr–MOF with the soc-type topological network, addresses several challenges related to the development of a highly efficient and robust heterogeneous catalyst for peptide-bond hydrolysis. An increase of more than three orders of magnitude was observed in the rate of the peptide-bond hydrolysis in a model dipeptide, highlighting the importance of the 6-connectivity of $Zr_6$ oxo-cluster sites in the MIP-201 framework for catalytic efficiency. Compared with few other MOFs, the most striking advantage of MIP-201 in catalyzing peptide-bond hydrolysis is the possibility to use catalyst under a wide range of reaction conditions, without compromising its stability. MIP-201 was shown to selectively hydrolyze myoglobin, which has 153 amino acids in its sequence, by cleaving the protein at only five peptide bonds. The excellent chemical and architectural stability of MIP-201 confirmed in the experiments with the protein, in addition to its high catalytic activity and recycling ability, further highlights its potential as an artificial protease. Furthermore, the green, scalable, and cost-effective synthesis of MIP-201 with a good product yield makes it a promising and practical alternative for selective hydrolysis of proteins in proteomics and biotechnology applications.

## Methods
**Synthesis of MIP-201**. $Zr(SO_4)_2 \cdot 4H_2O$ (10.8 g) and $H_4$mdip (5.1 g) were transferred to a round-bottom flask (1 L), followed by the addition of water (300 mL)

and acetic acid (100 mL) under stirring at room temperature. The reaction was refluxed at 120 °C for 12 h and cooled to R.T. The resulting solid product was collected by filtration, washed with ethanol, and air-dried. Crude product with a light-gold color (12.5 g, 90% yield based on H₄mdip) was obtained.

**Peptide-bond hydrolysis studies**. The hydrolysis reactions were studied at pD 7.4 and 60 °C by applying a general procedure. $D_2O$ (950 μL) was added to a solid sample of MIP-201 (2.8 mg, 2.0 μmol) in a glass vial followed by stirring for 30 min at room temperature. Peptide (50 μL of 40 mM stock solution, 2.0 μmol) was added to the suspension and the pH of reaction mixture was adjusted to pD 7.4 by using NaOD. The samples were incubated at 60 °C. After different time increments, the reaction mixture was centrifuged at 15,000 rpm for 20 min to remove the MOF. The $^1H$ NMR spectra of the resulting solutions were recorded using TMSP-d4 as internal reference. The rate constants were obtained by fitting peptide concentrations at different time increments to a first-order decay function.

**Protein-hydrolysis studies**. Mb (0.02 mM) was mixed with 2.8 mg of MIP-201 in HEPES buffer at pH 7.4 or in pure water at pH 7.4. Samples were stirred at 60 °C and aliquots for SDS-PAGE analysis were taken at different time increments. Stacking gel was 4% (w/v) polyacrylamide in 0.5 M Tris-HCl buffer at pH 6.8 and resolving gel consisted of 18% (w/v) polyacrylamide in 1.5 M Tris-HCl buffer at pH 8.8. Sample buffer (5 μL) was added to 15 μL of the reaction mixture, and after heating at 100 °C for 5 min, 10 μL of the resulting solution was loaded on the gel. Molecular-mass standard ladder used in SDS-PAGE was Page Ruler unstained low-range protein. An OmniPAGE electrophoretic cell was combined with an EV243 power supply and the experiments were performed at 200 V for 1.5 h.

**Catalyst-recycling experiments**. Recyclability of MIP-201 was tested by repeating Gly–Gly hydrolysis five times starting from one batch of catalyst in a glass vial. After each run, the reaction mixture was centrifuged and the solution was used to analyze the amount of unreacted Gly–Gly and the hydrolyzed products (Gly and cyclic Gly–Gly) by $^1H$ NMR. Water was added to the solid MOF and stirred for one hour before removing it. This process was repeated four times. The MOF material was subsequently stirred in an organic solvent (chloroform, ethanol, methanol…) for one day to exchange water. The procedure was repeated twice and the recycled MIP-201 was air-dried and activated at 120 °C for four hours before being used for the next catalytic run.

## Data availability

All data involved in this work are included in this article and the corresponding supplementary information. They are available from the corresponding authors upon reasonable request. The crystal-structure data have been deposited at CCDC under the deposition numbers CCDC: 2076200 and 2076201. These data can be obtained free of charge from the CCDC database via www.ccdc.cam.ac.uk.

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

## Acknowledgements

The authors from France acknowledge the ANR Project MeaCoPA (ANR-17-CE29-0003) and the European Community Seventh Program (FP7/2007–2013) for funding the research under Grant Agreement 608490 (Project M4CO2). S.W. acknowledges support from the National Natural Science Foundation of China (22071234) and the Fundamental Research Funds for the Central Universities (WK2480000007). T.P.V., C. Simms and H.G.L.T. thank Research Foundation Flanders (FWO) for funding (68090/11C9320N). T.P.V. thanks the European Commission (Horizon 2020, FoodEnTwin project, GA No. 810752) for funding. The computational work was granted access to the HPC resources of CINES under the allocation A0100907613 made by GENCI.

## Author contributions

Conceptualization, S.W., HGT.L., T.P.V., and C.S.; Investigation, S.W., HGT.L., M.W., C. Simms, I.D., A.T., G.M., T.P.V., and C.S.; Writing—original draft, S.W.; Writing—review & editing, S.W., HGT.L., M.W., G.M., T.P.V., and C.S. Supervision, T.P.V. and C.S.

## Competing interests

The authors declare no competing interests.
