## [Peer Review File · Nature Communications]

Title: A Zirconium Metal-Organic Framework with SOC Topological Net for Catalytic Peptide Bond HydrolysisREVIEWER COMMENTS

Reviewer #1 (Remarks to the Author):

The study is comprehensive, and the experimental procedures were performed competently. The novel POM was described in high detail and indeed has promising properties. My questions regard the reaction specificity, selection of reaction substrates and reaction conditions.

1. From the study of myoglobin the authors concluded that the reaction occurred solely after Asp. This is an important proposition and should be confirmed beyond doubt, even if the other systems behaved similarly. An estimate based on gel band positions is not sufficient. The detection of peptides by mass spectrometry should be obtained.
2. Assuming the Asp-Xaa specificity in the protein, logically the Asp-Gly-amide should be used as dipeptide model rather than Gly-Gly, or at least in addition to Gly-Gly. I understand that a carboxylate in vicinity of the peptide bond is present in both cases, but spatially they are quite different, and using the more appropriate dipeptide model would allow for a quantitative comparison.
3. The reaction rate and yield are moderate at 60°C, which is a high temperature for the work with biological materials, as it would lead to prompt protein denaturation and likely aggregation. What is the temperature dependence of the yield?
4. MIP-201 seems to degrade with respect to its catalytic abilities after a few reaction cycles in water, the most important of the tested solvents. Is there a way to increase its durability?

Reviewer #2 (Remarks to the Author):

This manuscript reports a novel Zr-based MOF with potential application as catalyst in peptide hydrolysis. The new Zr-MOF has been thoughtfully characterized and a precise structural definition has been obtained with an interesting combination of experimental PXRD and computational data. To test the catalytic performance of MIP-201, the authors initially selected the model Gly-Gly dipeptide. In this case, MIP-201 outperforms in terms of activity and of robustness at different conditions (including a broad range of pHs) to other previously reported Zr-MOF and Zr-polyoxometalate catalysts. While the discussion and comparison of catalyst activity for dipeptide is clear and well-supported with kinetic data, in the case of Mb protein I missed a clear discussion assessment of the activity of catalyst MIP-201 and a comparison with the previously reported performance. I also missed some discussion of the mechanism that may be related to the previous point. For Mb protein I assume that MIP-201 acts as a surface catalyst, and it would be interesting to discuss which Zr sites can be responsible for the activity. For Gly-Gly dipeptide, it is not clear whether the dipeptide activation occurs on the surface or into the pores. Is the latter situation responsible for the high hydrolysis activity observed for dipeptide? In my opinion, the above-mentioned issues should be clarified before the publication is accepted. If that is the case, I would qualify this work as a remarkable achievement in the field of protein engineering by artificial proteases that merits publication in *Nat. Commun.*

Regarding the formal aspects, the manuscript is well-written and concise, and the graphical material is

comprehensive. My only comment is that the authors should proofread the manuscript and unify the format for compounds label (sometimes bold, sometimes not).

Reviewer #3 (Remarks to the Author):

In this submission, Serre et al reported a microporous MIP-201, which represents the first Zr-MOF with the soc type topological network and can be used as a highly efficient and robust solid catalyst for peptide bond hydrolysis. Compared to few other MOFs, the most striking advantage of MIP-201 in catalyzing peptide bond hydrolysis is the possibility to use catalyst under wide range of reaction conditions, without compromising its stability. In particular, MIP-201 370 was capable of selectively hydrolyzing myoglobin with 153 amino acids in its sequence, selectively cleaving the protein at only four peptide bonds. This work is very interesting and can be published in Nature Communication after addressing the following issues.

- 1.The authors think the high stability and catalytic active are due to the 6-connected Zr₆ cluster. I am wondering if the other reported 6-connected Zr₆-MOF have such high performance in catalyzing peptide bond hydrolysis.
- 2.Can the authors give the mechanism of the MIP-201-catalyzed reaction in SI

Reviewer #1 (Remarks to the Author):

The study is comprehensive, and the experimental procedures were performed competently. The novel POM was described in high detail and indeed has promising properties. My questions regard the reaction specificity, selection of reaction substrates and reaction conditions.

We thank the referee for the support and positive valuation of our work.

Question: From the study of myoglobin the authors concluded that the reaction occurred solely after Asp. This is an important proposition and should be confirmed beyond doubt, even if the other systems behaved similarly. An estimate based on gel band positions is not sufficient. The detection of peptides by mass spectrometry should be obtained.

Answer: We thank the referee for this suggestion, and agree that specificity of the MOF material in cleaving peptide bonds is an important issue that needs to be further investigated, and that mass spectrometric analysis is probably the best technique to provide such information. However, while SDS PAGE analysis reported in this work was done on the mixtures which contains MOF and proteolytic fragments, the mass spectrometric analysis requires complete removal of the MOF from the mixture. We observed that the removal of the MOF results in significant loss of proteins fragments, either due to their absorption on the MOF external surface or their inclusion into MOF pores. This in turn leads to incomplete coverage of the fragments that are resulting from protein hydrolysis. Digesting MOFs on the other hand also could result in further cleavage of the fragments due to the harsh condition used, again complicating mass spectrometric analysis. We would like to point that such issues were not present for Hf₁₈ cluster used in our previous work (Angew. Chem. Int. Ed., 2020, 59, 9094-9101), due to its discrete nature and therefore we were able to perform MS analysis of digestion products. We are currently pursuing the development of a protocol for mass spectrometric analysis by probing different ways to desorb protein fragments from the MOF.

The focus of the current study is not on the Asp-X specificity of MIP-201 but rather on the fact that the new catalyst exhibits remarkable reactivity and stability compared to other Zr-MOFs, and that it produces fragments with MW which are suitable for the middle down proteomics applications. Please note that in the original text of the paper, we did not state that the MIP-201 exhibits Asp-X specificity but that SDS-PAGE data “suggest that Mb was hydrolyzed at Asp-X peptide bonds, in accordance with the previously reported affinity of Zr(IV) based artificial proteases to preferentially cleave proteins next to Asp residues”. In order to be clearer that this is currently only a hypothesis we modified the text in the revised manuscript as following:

“While the exact position of the cleavage sites has not been determined at this point, the bands observed at 60 °C give indication that Mb was likely hydrolysed in the vicinity of Asp20-Ile21 (14.9 kDa), Asp122-Phe123 (13.6 kDa), Asp109-Ala110 (12.5 kDa), and Asp60-Leu61 (10.1 kDa), while the bands at 11.0 and 7.1 kDa likely originated from the cleavage of Mb at two sites (at Asp20-Ile21 and Asp122-Phe123 for the band at 11.0 kDa; Asp60-Leu61 and Asp126-Asp127 for 7.1 kDa). At 37 °C the hydrolysis most likely occurred at Asp109-Ala110 (12.5 kDa) while the bands at 11.2 and 7.1 kDa likely originated from the cleavage of Mb at two sites as described above (at Asp20-Ile21 and Asp122-Phe123 producing the band at 11.2 kDa; Asp60-Leu61 and Asp126-Asp127 for 7.1 kDa). The position of these bands was in accordance with the previously reported affinity of Zr(IV) based artificial proteases to preferentially cleave proteins next to Asp residues. The hydrolysis of these Asp-X peptide bonds in Mb has been previously observed in the presence of Zr-POMs; however, those reactions were much slower, with the observable hydrolysis occurring only after 48 hours at pH=5.0. We are currently working on the

development of a protocol for mass spectrometric analysis which will allow unambiguous detection of the cleavage sites.” (line 370-387 in the revised main text).

Question: Assuming the Asp-Xaa specificity in the protein, logically the Asp-Gly-amide should be used as dipeptide model rather than Gly-Gly, or at least in addition to Gly-Gly. I understand that a carboxylate in vicinity of the peptide bond is present in both cases, but spatially they are quite different, and using the more appropriate dipeptide model would allow for a quantitative comparison.

Answer: We thank the referee for this suggestion. The referee is correct that in view of possible Asp specificity of MIP-201 material, Asp containing peptides might be more suitable model peptide than Gly-Gly. The Gly-Gly is frequently used as a model system for newly developed catalysts in order to have a direct comparison of its catalytic activity towards the peptide bond, without interference of other structural factors, which may either promote or inhibit hydrolysis.

We have performed additional experiments using Asp-Gly dipeptide as suggested by the referee. The corresponding results are now included in the revised manuscript (line 388-397), and are in accordance with our previous works with different Zr-MOFs (MOF-808, UiO-66-NH₂, UiO-66-NO₂, UiO-66-TFA, see *J. Am. Chem. Soc.* 2018, 140, 6325 and *ACS Appl. Nano. Mat.* 2020, 3, 8931 for example), which demonstrated hydrolytic activity of MIP-201 towards a broad range of dipeptides, including the Asp containing ones. These works have shown that Zr-MOFs were able to efficiently hydrolyze all examined dipeptides, and that the rates of hydrolysis were dependent on the chemical nature and the size of the side chain.

Question: The reaction rate and yield are moderate at 60 °C, which is a high temperature for the work with biological materials, as it would lead to prompt protein denaturation and likely aggregation. What is the temperature dependence of the yield?

Answer: We thank the referee for raising this point. We have performed additional experiments at 37 °C, and these new experiments revealed some interesting effects of temperature on the Mb hydrolysis by MIP-201. While the high temperatures needed to observe peptide bond hydrolysis are required due to the extreme kinetic inertness of the peptide bond, they also cause partial unfolding of the protein, as referee suggests. Our new results show that this is actually beneficial for the cleavage reaction as more fragments are observed at 60 °C than at 37 °C, because the unfolding is likely to make peptide bonds more accessible for interaction with the catalyst. Furthermore, the temperature also affects adsorption of the protein and its fragments onto MOF material, as prolonged incubation times at 37 °C resulted in the loss of both intact protein and the resulting fragments of hydrolysis. These data point out that hydrolysis at 60 °C is more practical, but they also indicate that the temperature could be used to tune the extent of protein hydrolysis and the number of resulting fragments. These new data are now discussed in the revised manuscript (line 354-370).

Question: MIP-201 seems to degrade with respect to its catalytic abilities after a few reaction cycles in water, the most important of the tested solvents. Is there a way to increase its durability?

Answer: We thank the referee for raising this point and apologize for the related ambiguity in the manuscript. As mentioned in the main text (line 298-306), a gradual decrease of MIP-201 catalytic activity was observed in the first four runs in cycling test. And a comparable efficiency was observed in the fifth run of catalysis (53%) in comparison with that of the fourth run (55%). These observations suggested that a gradual decrease of catalytically active sites took place and the number of the accessible active site eventually achieved an equilibrium. Therefore, we mentioned in the main text that it is likely

due to the microporous nature of the MIP-201 structure, in which partial pore blocking by guest molecules (such as coordinated peptide, amino acid, water molecules) could lead to the decrease of catalytically active sites before it reached the equilibrium. The catalytic activity of the cycled MIP-201 sample should be kept when the number of the active site reached the equilibrium, starting from the fourth run in this case.

In order to further support the above conclusion, we have carried out extra experiments as detailed below. The catalytic activity of water-regenerated MIP-201 was checked for the sixth cycling run, and a comparable catalysis efficiency (52%) was observed in comparison with those of the fourth and fifth runs. It showed that the catalytic activity of cycled MIP-201 sample could be kept very well after the fourth run. In addition, we have collected PXRD data on the cycled MIP-201 samples. No notable difference could be observed between the PXRD patterns of the fresh and cycled MIP-201 samples, which support that the crystalline structure of MIP-201 was stable in the cycling test.

Therefore, the excellent chemical stability and constant catalytic performance of the equilibrated MIP-201 sample are good support for its reliable durability. We would like to point that increasing temperature and extending duration in catalyst activation could be possible ways to remove those pore-blocking residues in the cycled MIP-201 catalyst, and thus expose adequate active sites to approach the similar efficiency in each catalytic cycle.

Reviewer #2 (Remarks to the Author):

This manuscript reports a novel Zr-based MOF with potential application as catalyst in peptide hydrolysis. The new Zr-MOF has been thoughtfully characterized and a precise structural definition has been obtained with an interesting combination of experimental PXRD and computational data. To test the catalytic performance of MIP-201, the authors initially selected the model Gly-Gly dipeptide. In this case, MIP-201 outperforms in terms of activity and of robustness at different conditions (including a broad range of pHs) to other previously reported Zr-MOF and Zr-polyoxometalate catalysts. While the discussion and comparison of catalyst activity for dipeptide is clear and well-supported with kinetic data, in the case Mb protein I missed in the a clear discussion assessment of the activity of catalyst MIP-201 and a comparison with the previously reported performance. I also missed some discussion of the mechanism that may be related to previous point. For Mb protein I assume that MIP-201 acts as a surface catalyst, and it would interesting to discuss which Zr sites can be responsible of the activity. For Gly-Gly dipeptide, it is not clear whether the dipeptide activation occurs in the surface or into the pores. Is the latter situation responsible of the high hydrolysis activity observed for dipeptide? In my opinion, the above mentioned issues should be clarified before the publication is accepted. If that is the case, I would qualify this work as a remarkable achievement in the field of protein engineering by artificial proteases that merits publication in Nat. Commun.

Answer: We thank the referee for the positive evaluation of our work, and for their useful suggestions. The referee raises indeed an important question related to the activation of the peptide and protein substrates by MOFs. Our previous data with Zr-MOFs evaluated so far suggest that activation of small peptides may occur both within the pores and on the outer surface of Zr-MOFs. For example, we previously evaluated possible changes of NU-1000 surface and pores during and after the Gly-Gly hydrolysis reaction by performing N₂ sorption with MOF recovered from the GG hydrolysis reactions (see *Chem. Sci.* 2020, 11, 6662). While only a slight reduction was observed in the BET surface area before and after reaction (2192 and 2161 m² g⁻¹ respectively), a ≈15% reduction of the largest pore's

diameter from 34 to 29 Å was observed after incubation with Gly-Gly, evidencing that the pores are at least partially filled after reaction with Gly-Gly. Accordingly, small amounts of peptide and amino acid products, were detected through ¹H NMR analysis of a digested sample of NU-1000 recovered after reaction. These results were also in accordance with the linear trend observed in the Arrhenius plot, which is characteristic for a reaction without diffusion limitations, and is consistent with the freedom observed of Gly-Gly to enter and leave the pores in the adsorption experiments.

Regarding protein hydrolysis, our previous studies indicated nearly complete adsorption of proteins on Zr-MOF external surface. In accordance with the absence of protein in solution, N₂ physisorption measurements of a NU-1000 sample recovered after the reaction with the protein showed a reduction of the BET surface area from 2192 to 1612 m² g⁻¹, while no significant changes in the pore sizes were observed. These results indicate indeed significant absorption of protein onto the surface and not in the pores of MOF.

In order to clarify this, and following the referee's suggestion we have added the following text in the revised manuscript:

“These Zr(IV) sites are present on the outer surface and in the pores of MIP-201, and in principle both sites could lead to the activation of Gly-Gly. Previous studies of N₂ sorption isotherms with NU-1000 revealed slight reduction in the BET area before and after reaction and ≈15 % reduction of the largest pore's diameter after incubation with Gly-Gly, evidencing that the pores are at least partially filled after reaction with the peptide.” (line 265-271 in the revised manuscript)

And

“Considering the structure of MIP-201 and the size of Mb protein, it is very likely that the hydrolysis of Mb occurs on the outer surface of the particles. A previous study with NU-1000 revealed very little changes in the MOF pore size upon incubation with the protein, while significant changes in the BET area were detected, indicating significant absorption of protein onto the surface and not in the pores of MOF.” (line 404-409 in the revised manuscript)

Question: Regarding the formal aspects, the manuscript is well-written and concise, and the graphical material is comprehensive. My only comment is that the authors should proofread the manuscript and unify the format for compounds label (sometimes bold, sometimes not).

Answer: We thank the referee for this suggestion. We have unified the format of compounds label in the revised manuscript. And the revised manuscript was proofread by a native speaker.

Reviewer #3 (Remarks to the Author):

In this submission, Serre et al reported a microporous MIP-201, which represents the first Zr-MOF with the soc type topological network and can be used as a highly efficient and robust solid catalyst for peptide bond hydrolysis. Compared to few other MOFs, the most striking advantage of MIP-201 in catalyzing peptide bond hydrolysis is the possibility to use catalyst under wide range of reaction conditions, without compromising its stability. In particular, MIP-201 was capable of selectively hydrolyzing myoglobin with 153 amino acids in its sequence, selectively cleaving the protein at only

four peptide bonds. This work is very interesting and can be published in Nature Communication after addressing the following issues.

We thank the referee for the support and positive valuation of our work.

Question: The authors think the high stability and catalytic active are due to the 6-connected Zr_6 cluster. I am wondering if the other reported 6-connected Zr_6 -MOF have such high performance in catalyzing peptide bond hydrolysis.

Answer: The referee raises an important point. We have previously shown that MOF-808, which also has a 6-connected Zr_6 cluster, is also very effective in hydrolyzing a peptide bond in short dipeptides and in hen egg white lysozyme protein (see *J. Am. Chem. Soc.* 2018, 140, 6325). However, as also acknowledged by the referee, the MIP-201 reported in this work was shown to be much more stable in a wide range of different pH values, and importantly, showed an exceptional recycling ability associated with easy regeneration process. Further proofs that the 6-connected Zr_6 cluster is more active towards hydrolysis of peptide bond compared to other Zr-MOFs with higher connectivity have been reported in our other recent works which evaluated hydrolytic activity of NU-1000 and UiO-66 MOFs (see for example *Chem. Sci.* 2020, 11, 6662, and *ACS Appl. Nano Mater.* 2020, 3, 8931-8938). These MOFs showed much lower reactivity towards peptide bonds, presumably due to the higher connectivity of Zr_6 clusters, which leaves less available sites on Zr(IV) for interaction with the substrate. In order to clarify this, and following referee's suggestion the revised version now states the following:

“This highlights the important role of 6-connection of the Zr_6 -oxo node on the catalytic activity and how critical the active site density is for the efficiency of a MOF artificial peptidase, since other Zr-MOFs with larger cluster connectivity exhibit lower catalytic activities.” (line 250-253 in the revised manuscript).

Question: Can the authors give the mechanism of the MIP-201-catalyzed reaction in SI

Answer: We thank the referee for raising this point. The mechanism of peptide bond hydrolysis by a related Zr-MOF, which also has 6-connected Zr_6 cluster has been evaluated in detail by DFT methods. In this work (see *Phys. Chem. Chem. Phys.*, 2020, 22, 25136) we revealed the key role of Zr_6 cluster in the activation of the peptide bond. In the revised manuscript, we refer to this mechanism by stating the following:

“Detailed density functional theory (DFT) calculations on the peptide bond hydrolysis by MOF-808 previously confirmed the central catalytic role of $[Zr_6O_8]$ cluster, and revealed that reaction is facilitated by peptide coordination to two adjacent Zr(IV) centres in the cluster. Such activation by two Zr(IV) centres likely explains the superior reactivity of Zr-MOF materials with respect to Zr(IV) polyoxometalate complexes, in which only a single Zr(IV) is present for peptide bond activation.” (line 254-260 in the revised manuscript).

REVIEWERS' COMMENTS

Reviewer #1 (Remarks to the Author):

I find the responses to the review provided by the authors to be comprehensive and fully satisfactory. I recommend publication of the revised version.

Reviewer #2 (Remarks to the Author):

The authors have satisfactorily answer to all my previous concerns and therefore I can support publication of the manuscript in Nat. Commun.

Reviewer #3 (Remarks to the Author):

The revised manuscript can be published in its current form.

REVIEWERS' COMMENTS

Reviewer #1

I find the responses to the review provided by the authors to be comprehensive and fully satisfactory. I recommend publication of the revised version.

Response: We thank the referee for the positive feedback and the support for the publication of our paper.

Reviewer #2

The authors have satisfactorily answer to all my previous concerns and therefore I can support publication of the manuscript in Nat. Commun.

Response: We thank the referee for the positive feedback and the support for the publication of our paper.

Reviewer #3

The revised manuscript can be published in its current form.

Response: We thank the referee for the positive feedback and the support for the publication of our paper.